# Utility of Clinical Next Generation Sequencing Tests in KIT/PDGFRA/SDH Wild-Type Gastrointestinal Stromal Tumors

**DOI:** 10.3390/cancers16091707

**Published:** 2024-04-27

**Authors:** Ryan A. Denu, Cissimol P. Joseph, Elizabeth S. Urquiola, Precious S. Byrd, Richard K. Yang, Ravin Ratan, Maria Alejandra Zarzour, Anthony P. Conley, Dejka M. Araujo, Vinod Ravi, Elise F. Nassif Haddad, Michael S. Nakazawa, Shreyaskumar Patel, Wei-Lien Wang, Alexander J. Lazar, Neeta Somaiah

**Affiliations:** 1Division of Cancer Medicine, The University of Texas MD Anderson Cancer Center, Houston, TX 77030, USA; 2Department of Sarcoma Medical Oncology, The University of Texas MD Anderson Cancer Center, Houston, TX 77030, USA; 3Department of Pathology, Division of Pathology & Laboratory Medicine, The University of Texas MD Anderson Cancer Center, Houston, TX 77030, USA

**Keywords:** GIST, genomics, sarcoma, wild type GIST, next generation sequencing

## Abstract

**Simple Summary:**

Most gastrointestinal stromal tumors (GISTs) are driven by activating mutations in *KIT* and *PDGFRA* or alterations in the succinate dehydrogenase (SDH) complex. A small fraction of GISTs lack alterations in *KIT*, *PDGFRA*, and the *SDH* complex, so-called “triple-negative” GISTs. We assessed clinical genomic sequencing, treatment, and survival outcomes in a cohort of 20 triple-negative GISTs. Genomic alterations were most commonly seen in the RAS/RAF/MAPK pathway and the DNA damage response pathway. Compared to *KIT*/*PDGFRA* mutant GIST, limited benefit was observed with imatinib in triple-negative GIST. In-depth molecular profiling can be helpful in identifying driver mutations and guiding therapy.

**Abstract:**

**Objective:** The vast majority of gastrointestinal stromal tumors (GISTs) are driven by activating mutations in *KIT*, *PDGFRA*, or components of the succinate dehydrogenase (SDH) complex (*SDHA*, *SDHB*, *SDHC*, and *SDHD* genes). A small fraction of GISTs lack alterations in *KIT*, *PDGFRA*, and *SDH*. We aimed to further characterize the clinical and genomic characteristics of these so-called “triple-negative” GISTs. **Methods:** We extracted clinical and genomic data from patients seen at MD Anderson Cancer Center with a diagnosis of GIST and available clinical next generation sequencing data to identify “triple-negative” patients. **Results:** Of the 20 patients identified, 11 (55.0%) had gastric, 8 (40.0%) had small intestinal, and 1 (5.0%) had rectal primary sites. In total, 18 patients (90.0%) eventually developed recurrent or metastatic disease, and 8 of these presented with de novo metastatic disease. For the 13 patients with evaluable response to imatinib (e.g., neoadjuvant treatment or for recurrent/metastatic disease), the median PFS with imatinib was 4.4 months (range 0.5–191.8 months). Outcomes varied widely, as some patients rapidly developed progressive disease while others had more indolent disease. Regarding potential genomic drivers, four patients were found to have alterations in the RAS/RAF/MAPK pathway: two with a *BRAF* V600E mutation and two with *NF1* loss-of-function (LOF) mutations (one deletion and one splice site mutation). In addition, we identified two with *TP53* LOF mutations, one with *NTRK3* fusion (*ETV6-NTRK3*), one with *PTEN* deletion, one with *FGFR1* gain-of-function (GOF) mutation (K654E), one with *CHEK2* LOF mutation (T367fs*), one with Aurora kinase A fusion (*AURKA-CSTF1*), and one with *FANCA* deletion. Patients had better responses with molecularly targeted therapies than with imatinib. **Conclusions:** Triple-negative GISTs comprise a diverse cohort with different driver mutations. Compared to *KIT*/*PDGFRA*-mutant GIST, limited benefit was observed with imatinib in triple-negative GIST. In depth molecular profiling can be helpful in identifying driver mutations and guiding therapy.

## 1. Introduction

Gastrointestinal stromal tumors (GISTs) are the most common mesenchymal neoplasms of the gastrointestinal tract and arise from the interstitial cells of Cajal [1]; however they are relatively uncommon, with an estimated incidence of approximately 7 cases per million people per year in the United States [2]. The vast majority of GISTs are driven by activating mutations in *KIT* or *PDGFRA* [3]. About 10–15% do not have an alteration in one of these two genes, and they are generally less sensitive to currently approved tyrosine kinase inhibitors (TKIs) [4,5,6]. Previous reports have some shed light on alternate oncogenic drivers in this subset of so-called “wild-type” GIST that are more prevalent in young females and associated with multifocal but indolent disease, gastric primary, and lack of response to traditional *KIT*/*PDGFRA* TKIs such as imatinib [7].

The next most common alterations after *KIT* and *PDGFRA* are in *SDHA*, *SDHB*, *SDHC*, and *SDHD* genes, components of the succinate dehydrogenase (SDH) complex that catalyzes the oxidation of succinate to fumarate in the tricarboxylic acid cycle [8,9,10]. SDH deficient GIST accounts for 7–10% of all GIST [11,12]. SDH deficiency leads to succinate accumulation, which inhibits prolyl hydroxylases, leading to accumulation of the hypoxia-inducible factor 1 alpha (HIF-1α) [13]. SDH-deficient GISTs are more commonly seen in pediatric patients and can be associated with either the Carney triad (GIST, paraganglioma, pulmonary chondroma) characterized by epigenetic silencing of the *SDHC* gene by the promoter hypermethylation or Carney–Stratakis syndrome (GIST and paraganglioma), which are caused by germline mutations in *SDH* [14,15,16,17].

There have been small case series and case reports of GIST lacking alterations in *KIT*, *PDGFRA*, and *SDH* (“triple-negative GIST”). One of the more common alterations in this cohort is in the RAS/RAF/MAPK pathway [18,19,20,21]. For example, some have *BRAF* mutations, which are often the classic V600E gain-of-function (GOF) mutation that is seen frequently in melanoma and other cancer types [22,23], and this is often mutually exclusive with *KIT*/*PDGFRA* mutations [24]. The next-most-common alteration is loss-of-function (LOF) mutations of *NF1* [18,20,21,25], a GTPase-activating protein that promotes the hydrolysis of Ras-bound GTP, thereby acting as a negative regulator of the RAS/RAF/MAPK pathway; *NF1* LOF mutation results in RAS/RAF/MAPK pathway activation. Alterations in DNA damage repair genes have also been reported [21]. Other studies have shown that about 20% of tumors thought to be *KIT*/*PDGFRA*/*SDH* wild-type end up having cryptic pathogenic *KIT* mutations [20]. However, most of these studies to date have used targeted sequencing of a limited number of genes, and how these triple-negative GISTs behave clinically is not well understood. In this study, we aimed to characterize the genomic landscape of triple-negative GIST using clinical next generation sequencing with larger gene panels and the associated treatment and outcome data.

## 2. Materials and Methods

### 2.1. Study Design

This is a retrospective study of all patients treated at MD Anderson between 1 January 2015 and 31 December 2023 with a diagnosis of GIST and available clinical next-generation sequencing data. We included those that lacked alterations in *KIT* and *PDGFRA* and either (1) retained expression of *SDHB* by immunohistochemistry (15 patients) or (2) had wild-type SDHB according to a sequencing assay (5 patients had only SDH mutation testing and not SDHB immunohistochemistry). The following clinical data were collected by retrospective chart review: age, gender, race, ethnicity, date of birth, date of diagnosis, vital status, tumor grade, tumor mitotic rate, tumor stage, surgery date, systemic therapies and dates of treatment, response to treatment, personal history of cancer, family history of cancer, and results of next generation sequencing tests. Response to treatment was based on the treating clinician’s documentation in the patients’ charts.

### 2.2. Clinical Genomic Sequencing

Clinical sequencing was available from multiple platforms. Twelve patients had sequencing with an MD Anderson platform. Five patients had BostonGene, four had FoundationOne^®^, one patient had Caris, one patient had NeoTYPE^®^, one patient had Endeavor, and two had outside institution panels. Five patients had multiple panels: one patient had BostonGene, FoundationOne^®^, and MD Anderson panels; one had FoundationOne^®^ and MD Anderson panels; one had Caris and BostonGene; one had NeoTYPE^®^ and BostonGene; and one had Endeavor and BostonGene. The MD Anderson platform has evolved to include more genes over time. The latest is the Mutation Analysis Precision Panel (MAPP, utilized for three patients) is an MD Anderson Molecular Diagnostic Laboratory (MDL)-developed and -validated Illumina hybrid capture-based assay which can detect mutations and sequence variants in 610 genes, copy number variants in 583 genes, select gene rearrangements in 34 genes, and select immune oncology signatures, including microsatellite instability (MSI) and tumor mutational burden (TMB); it compares these to matched non-tumor tissue to filter out single nucleotide polymorphisms and germline variants. The Solid Tumor Genomics Assay 2018 (STGA 2018, five patients) is an AmpliSeq chemistry-based ion torrent assay, which utilizes a germline control sample and was first implemented in 2018. It assesses mutations in 134 genes and selected copy number variations in 47 genes and compares to matched non-tumor tissue [26]. The CMS50 panel (four patients) also uses an AmpliSeq chemistry-based ion torrent assay, though it does not utilize a germline control. It was first implemented in 2012 and assessed 50 genes using PCR-based sequencing [27]. The MD Anderson Precision Oncology Decision Support (PODS) tool was used to assess for clinical actionability of genomic alterations [28].

### 2.3. Survival Analyses and Statistics

Statistical analyses were performed using GraphPad Prism (version 9.5.0 or higher, RRID:SCR_002798) and R (version 4.2.2 or higher).

Overall survival (OS) was calculated from the date of first histologic diagnosis (either pre-treatment biopsy or surgical pathology) to death or the latest follow-up. Recurrence-free survival (RFS) was calculated in patients with initially localized disease from the date of surgery of the primary tumor to the date of recurrence or the latest follow-up. Progression-free survival (PFS) was calculated in patients with metastatic disease from the start of therapy to the date of progression or the latest follow-up.

### 2.4. Data Availability

De-identified data generated in this study are available upon request from the corresponding author.

### 2.5. Ethics

This study was approved by the University of Texas MD Anderson Cancer Center Institutional Review Board (protocols 2022-0278 and LAB04-0890) and was conducted in accordance with the U.S. Common Rule. Clinical and genomic data were obtained following signed informed consent onto prospective institutional protocols or under retrospective review protocols with a limited waiver of authorization.

## 3. Results

### 3.1. Clinical Characteristics of Triple-Negative GIST Cohort

We identified 20 patients with GIST that lacked alterations in *KIT*, *PDGFRA*, and the SDH complex, so called “triple-negative GIST.” Patient characteristics are summarized in Table 1. Median age of diagnosis was 48 years (mean 45.3 years; Figure 1A), and twelve patients (60.0%) were female. The median tumor size was 7 cm (mean 9.6 cm; Figure 1B). The median mitotic index was 15 (per 50 hpf or 5 mm^2^), and the mean was 31.8 (Figure 1C). In total, 11 (55.0%) triple-negative GISTs were gastric, 8 (40.0%) were small intestinal, and 1 (5.0%) was rectal (Figure 1D); 8 patients (40.0%) presented with de novo metastatic disease (i.e., stage IV; Figure 1E); and 19 patients total (95.0%) eventually developed recurrent or metastatic disease versus 1 (5.0%) with localized disease at the time of most recent follow-up (median follow-up of 66.4 months from the time of diagnosis). Of the 19 with recurrent or metastatic disease, 8 had local relapse in the abdomen or pelvis, 9 had metastasis to the liver, 5 to the omentum or peritoneum, 1 to the pancreas, and 1 to the spleen.

### 3.2. The Genomics of Triple-Negative GIST

All 20 cases have had clinical genomic sequencing to reveal a potential driver (Table 1, Figure 2A). The mean and median number of somatic mutations identified were 2.0 and 1, respectively (range 0–8; Figure 2B). The most commonly altered genes were *NF1*, *BRAF*, *FAM123B*, and *PIK3CA* (Figure 2C). Regarding hypothesized drivers identified by sequencing (Figure 2D), four patients were found to have alterations in the RAS/RAF/MAPK pathway: two with a *BRAF* V600E GOF mutation, two with *NF1* LOF mutations (1 deletion and 1 splice site mutation), and two with *TP53* LOF mutations. However, one of the patients with a *TP53* mutation (patient 15) also had three other potential driver mutations: *NF2* c.517-1G > A (splice site mutation listed as pathogenic in ClinVar and predicted to result in loss of protein function) and amplification of *FGF3*, *FGF19*, and *CCND1* (all listed as likely oncogenic by OncoKB). There was one other case with a likely non-pathogenic *NF1* mutation (H2457R). In addition, one tumor each had a *NTRK3* fusion (*ETV6-NTRK3*), a *PTEN* deletion, an *FGFR1* GOF mutation (K654E), a *CHEK2* LOF mutation (T367fs*), and an Aurora kinase A fusion (*AURKA-CSTF1*). Two patients had *PIK3CA* mutations (both I391M), though this alteration has not been identified as a driver mutation. Sequencing of the remaining seven patients (45.0%) did not identify an obvious driver mutation.

### 3.3. Response to Treatment

Regarding treatment, 13 (65.0%) had surgery as the initial treatment, while 7 (35.0%) had medical therapy as the initial treatment. Of the eight patients with de novo metastatic disease, four initially started with imatinib and three developed progressive disease after 2.0, 4.2, and 4.4 months. One is still on imatinib 11 years after initial diagnosis. The remaining four patients with stage IV disease had metastatic disease discovered at the time of initial surgery by surgical pathology or were undergoing surgery for oligometastatic disease to render them disease-free.

All 20 patients were treated with imatinib, the most common first-line therapy for GIST, at some point in their treatment course. The mean number of lines of therapies was 3.0 (median 2, range 1–9). For the 13 patients with evaluable response to imatinib (e.g., neoadjuvant treatment or for recurrent/metastatic disease), median PFS from the start of therapy with imatinib was 4.4 months (range 0.5–129.4 months; Figure 3A).

The results of clinical sequencing testing led to treatment with a molecularly targeted treatment for which there is an FDA-approved drug in three patients (Figure 3A). Two patients with *BRAF* V600E mutation have both been treated with a combination of BRAF and MEK inhibitors. One patient has been on this therapy for 34.2 months, has had a partial response, and treatment is ongoing; unfortunately, the other patient had to stop therapy after two months due to toxicity. The patient with *ETV6-NTRK3* fusion was treated with larotrectinib for 28.7 months prior to progression, likely due to an acquired *NTRK3* gatekeeper mutation (F617L) rendering resistance to larotrectinib [29]. This patient was then transitioned to a second-generation NTRK inhibitor, selitrectinib, and was on this for 16.5 months prior to progression. In addition to these, Figure 3B displays the timeline of treatments for all patients in the cohort arranged by the purported driver.

### 3.4. Survival Outcomes

Follow-up data were available for a median of 66.4 months from the time of diagnosis, which was defined as the date of pathologic confirmation of GIST. Regarding clinical outcomes, the median OS was 301.2 months (Figure 4A). For the 11 patients initially treated with curative intent, the median recurrence-free survival was 57.4 months (Figure 4B). For the 10 patients with localized disease initially that later developed recurrent/metastatic disease, the median time to recurrence/metastasis from the initiation of treatment was 43.3 months (range 1.1–509.9 months, mean 105.4 months). For patients with advanced/metastatic disease, the median PFS with first-line systemic treatment was 23.1 months (Figure 4C). The median OS for patients that presented with de novo metastatic disease (i.e., stage IV) was 66.8 months, and the median OS for patients that ever developed metastatic disease was 301 months.

We next looked at whether the specific driver mutation was associated with differences in outcomes. The two patients in our cohort with *TP53* mutation had notably worse outcomes. One patient had OS of 3.7 and PFS of 1.1 months on imatinib, while the second patient had OS of 7.9 months and PFS of 3.8 months on imatinib. In the patients with the best outcomes (either no recurrence/progression or recurrence/progression 5 years or more after diagnosis, n = 8), there was no obvious association with genomic driver; one with *FGFR1* GOF mutation, one with *BRAF* GOF mutation, one with *CHEK2* LOF mutation, one with *FANCA* deletion, and the remaining four without identified drivers.

### 3.5. Review of Literature on Triple-Negative GIST

Lastly, we reviewed the literature for all cases of triple-negative GIST (Appendix A). Including the patients reported in our cohort, we identified a total of 112 cases. The mean and median ages of diagnosis were 54.7 and 56.5 years, respectively. There were 49 females (51.0% of cases with sex reported) and 47 males (48.9%). There was a predilection for small intestinal GIST (64.9% of cases with primary site listed) compared to gastric (22.3%), colorectal (4.3%), and peritoneal/retroperitoneal (5.3%). The mean and median tumor sizes were 7.4 cm and 6.5 cm, respectively. The mean and median mitotic rates were 23.1 and 8 per 50 hpf or 5 mm^2^ (range <5 to 160), respectively. *BRAF* mutations (33 cases, 29.5%), *NF1* LOF mutations (24 cases, 21.4%), and *FGFR1* pathway alterations (13 cases, 11.6%) were the most common alterations. Other reported alterations in more than one case included *NTRK3* fusion and *TP53* LOF mutation.

## 4. Discussion

In this study we identified 20 cases of so-called “triple-negative GIST”, which are those that are *KIT* WT, *PDGFRA* WT, and with intact SDH complex. Comparing patient characteristics of our triple-negative GIST cohort to all GISTs, our cohort was younger with an average age of 45 years versus 64 in other cohorts [30]. Our triple-negative cohort was more likely to present with stage IV disease (35% with stage IV versus 17% in other studies of all GIST) [30]. Regarding genomics, in our cohort 4 patients had an alteration in the RAS/RAF/MAPK pathway (2 with *BRAF* V600E mutation and 2 with *NF1* alterations). Two had *TP53* LOF mutations, though one of these also had additional potential driver mutations (*NF2* splice site mutation and amplification of *CCND1*, *FGF3*, and *FGF19*). Among the rest, one tumor each had a *ETV6-NTRK3* fusion, a *PTEN* deletion, an *FGFR1* gain-of-function mutation, a *CHEK2* LOF mutation (T367fs*), an *AURKA-CSTF1* fusion, and a *FANCA* deletion. The remaining 8 did not have an identifiable genomic driver. Following *KIT*, *PDGFRA*, and *SDH* alterations, the next most common alterations reported in the literature are RAS/RAF/MAPK pathway alterations, which is consistent with our findings. This pathway is most often altered by *BRAF* V600E mutation or *NF1* LOF [18,19,20,23,31,32,33,34,35]. The term “quadruple-negative GIST” has been used in the literature to describe GISTs that have wild-type *KIT* and *PDGFRA*, intact SDH complex, and unaltered RAS/RAF/MAPK pathway [18,36]. Of the 20 patients in our cohort, 16 (80%) could be described as “quadruple-negative”.

Outside of RAS/RAF/MAPK pathway alterations, one of the next-most-commonly altered pathways is the FGFR1 pathway [37]. Multiple studies have identified different alterations in this pathway. Some studies have identified GOF mutations in the *FGFR1* receptor itself [38,39], including the current study. *FGFR1* fusions have also been identified [39]. *FGF3*, *FGF4,* and *FGF19* are adjacent genes on chromosome 11q13, are frequently co-amplified and are known to be ligands for FGFR1 [40]. *FGF4* amplification has been reported in wild-type GIST [41], and our study identified *FGF3* and *FGF19* amplification in one case. Taken together, there appear to be multiple mechanisms leading to increased FGFR1 activity as a driver in a sizable fraction of quadruple-negative GISTs [37]. Interestingly, SDH-deficient GISTs have also been shown to upregulate *FGF4* expression via DNA-hypermethylation-mediated disruption of an insulator that normally separates a super-enhancer from the *FGF4* gene [42]. Concordantly, an SDH-deficient patient-derived xenograft model was sensitive to FGFR inhibition [42]. Interestingly, regorafenib is a TKI approved for GIST that also targets FGFR and has shown benefit in SDH-deficient GIST [43]. Therefore, the FGFR pathway appears to be important for the pathogenesis of wild-type GIST and may be an important therapeutic target.

There are a number of rarer drivers that have been identified. Our study and others have identified *NTRK3* fusions, which are readily targetable with NTRK TKIs such as entrectinib and larotrectinib [39,44]. One study identified a mutation in the E3 ubiquitin ligase *CBL*, a *KIT*-*PDGFRA* fusion, and an *ARID1A* mutation as potential drivers of wild-type GIST [19]. Another study performed whole exome sequencing of nine cases of quadruple-negative GIST, finding somatic oncogenic mutations in *TP53, MEN1, MAX, FGFR1, CHD4,* and *CTDNN2* [38]. Another study of two cases of quadruple-negative GIST used transcriptomics and found that these two cases had a distinct gene expression profile from other GISTs characterized by overexpression of *CALCRL*, *COL22A1*, *NTRK2*, *CDK6*, and *ERG* [36]. A final study of 72 quadruple-negative GISTs in China showed that 27.78% and 25% had *TP53* and *RB1* mutations, respectively [45]. There were also mutations identified in many other genes, including *ALK*, *CCNE1*, *MYC*, *PIK3CA*, *POLE*, and *PTEN*, among others, though it is unclear whether or not these represent driver mutations. Our review of the literature and analysis of all these aforementioned cases of triple-negative and quadruple-negative GIST suggests that *BRAF* V600E, *NF1* LOF mutations, and FGFR1 pathway GOF alterations are the most common drivers of triple-negative GIST.

In our study, advanced/metastatic triple-negative GIST was associated with better survival than other GISTs with an OS of 301.2 months (~25 years) compared to a contemporary overall survival of approximately 5 years for all GISTs who get multiple TKIs [46]. However, while most triple-negative GIST patients in our cohort had superior outcomes, there was a small number that had rapidly progressive disease, and one patient died from their disease 3.7 months after diagnosis. The two patients in our cohort with *TP53* mutation had notably worse outcomes (OS 3.7 and 7.9 months, PFS 1.1 and 3.8 months, respectively). It appears that triple-negative GIST patients generally have better survival outcomes compared to all GIST cases, though there is considerable heterogeneity.

Regarding treatment response to approved TKIs, our cohort demonstrated lower response rates than all GIST at-large. This is consistent with previous reports that have demonstrated lack of response to traditional *KIT*/*PDGFRA* TKIs in wild-type GIST [7]. Ours and prior studies highlight the importance of upfront sequencing to identify the driver, and if no alterations in *KIT* or *PDGFRA* are discovered to delve further with more comprehensive sequencing. There may also be benefit to comprehensive sequencing in *KIT*-mutant GIST after progression on imatinib, as some may have targetable resistance alterations. For example, *CDKN2A* deletion is a reported mechanism of resistance to imatinib and often co-occurs with deletion of an adjacent gene, *MTAP*, which may confer sensitivity to PRMT5 inhibitors and anti-folates [47,48,49]. Identification of *BRAF* mutations, one of the most common alterations in triple-negative GIST, is important to guide therapy. Based on data from the present study and others, *BRAF*-mutant GIST does not respond well to KIT TKIs but does respond well to combination BRAF and MEK inhibition, as is seen in other *BRAF*-driven cancers, such as melanoma. Another common alteration seen in triple-negative GIST is LOF mutation in *NF1*, which similarly serves to activate the RAS/RAF/MAPK pathway. An interesting future area of investigation will be to see if combination BRAF and MEK inhibition is effective in triple-negative GIST with *NF1* LOF mutation. Alterations in other pathways are much less common, but one worth mentioning is the DNA damage response pathway. In our cohort, we identified two patients with alterations in the DNA damage response genes *CHEK2* and *FANCA*. Two other studies have also identified alterations in DNA damage response genes in GIST [45,50]. It will be interesting to see if GISTs with alterations in DNA damage response genes respond to poly (ADP-ribose) polymerase (PARP) inhibitors. Lastly, given that *NTRK* fusions are occasionally seen in these triple-negative GISTs, and tumors harboring these fusions tend to respond well to NTRK inhibitors such as larotrectinib, it is important to utilize a sequencing assay with the ability to detect *NTRK* fusions.

This study has several limitations. First, this is a small retrospective analysis, and certain data, such as patient outcomes, may be prone to confounding and bias, which may make it difficult to generalize these findings to all wild-type GISTs. Second, we are bound by the limitations of the sequencing panels to detect driver mutations, and the panels were not completely overlapping, though common drivers identified were included in all the panels. The altered genes identified were not tested directly as specific drivers of the tumor but were classified as such based on data reported in the literature. However, the data from this cohort are comparable to what is reported in the literature and can serve as a guide to approaching this rare but challenging-to-treat population. Third, most but not all patients had SDHB immunohistochemistry performed, and we cannot exclude the possibility that those without SDHB immunohistochemistry may have had epigenetic silencing of the *SDHB* locus, which would not have been ascertained with sequencing. However, we identified driver mutations for most of these (one with *NTRK3* fusion, one with *BRAF* V600E mutation, and two with *TP53* mutation).

## 5. Conclusions

In summary, we report our experience with 20 patients with triple-negative GIST. Survival outcomes were superior but response to targeted TKIs in this cohort was inferior compared to *KIT*/*PDGFRA*-mutated GIST. Molecular profiling did reveal actionable alterations in a fraction of the triple-negative GIST cohort, and treatment with molecularly matched treatments resulted in greater clinical benefit than typical TKIs. In patients with no identifiable alterations on next-generation sequencing panels, a more comprehensive genetic and epigenetic evaluation might be worth pursuing to establish unidentified mechanisms of oncogenesis.

## Figures and Tables

**Figure 1 cancers-16-01707-f001:**
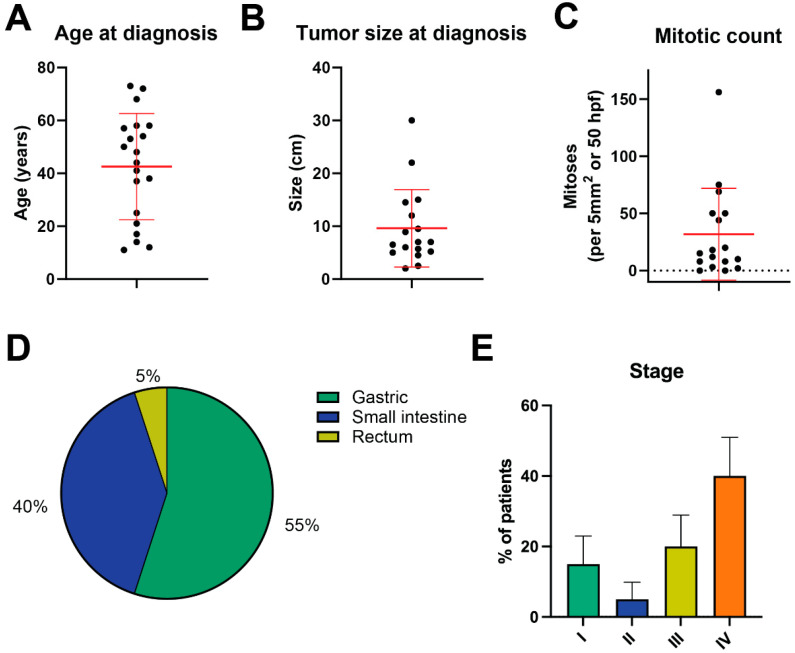
Clinical characteristics and outcomes in triple-negative GIST. (**A**) Distribution of age at diagnosis. (**B**) Distribution of tumor size at diagnosis. (**C**) Distribution of mitotic count from original biopsy or resected specimen. In (**A**–**C**), bars represent means ± SD. (**D**) Anatomic distribution of triple-negative GIST cases. (**E**) On the left in colored bars are the initial disease stage. Black and gray bars on the right indicate the percent of patients that remained with localized disease versus those that developed recurrent or metastatic disease. Bars represent percentages plus standard error of proportion.

**Figure 2 cancers-16-01707-f002:**
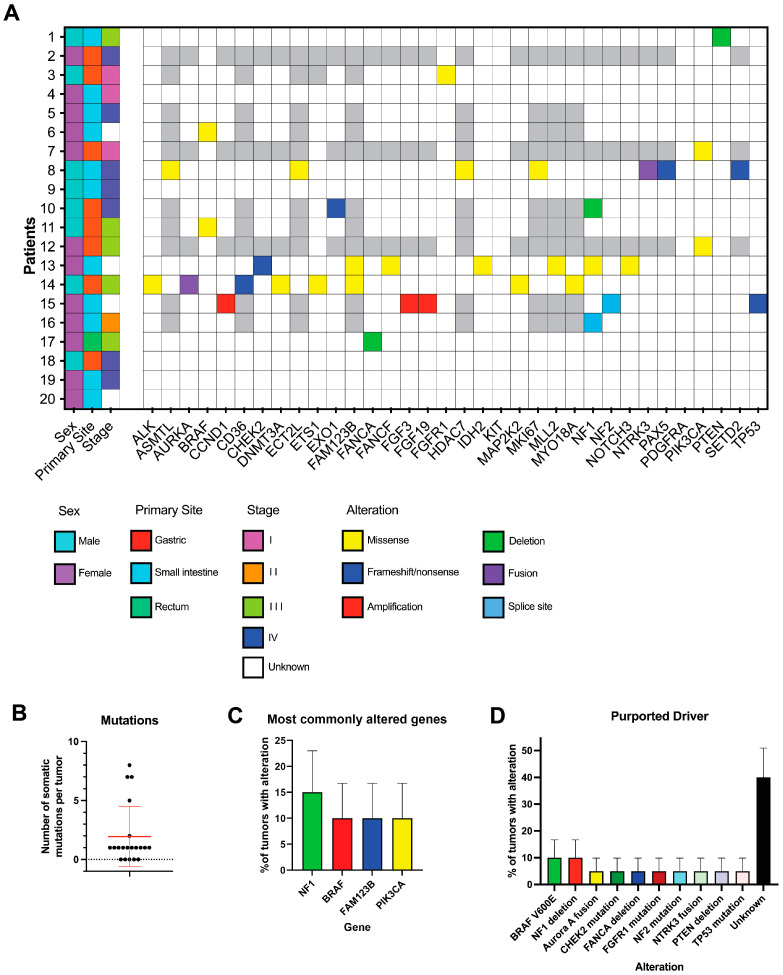
Genomics of triple-negative GIST. (**A**) Map of the genomic alterations in triple-negative GIST patients. Each row represents a patient. Each column represents a clinical feature (left 3 columns) or gene, as indicated. White boxes indicate that the gene was profiled but that no alteration was found, and gray boxes indicate that the gene was not profiled. (**B**) Number of somatic mutations detected by clinical sequencing assays. Each dot represents a single tumor, and bars represent mean ± SD. (**C**) Distribution of the most commonly altered genes in the cohort. (**D**) Percentage of tumors with each hypothesized driver mutation. In (**C**,**D**), percentages plus standard error of proportion are plotted.

**Figure 3 cancers-16-01707-f003:**
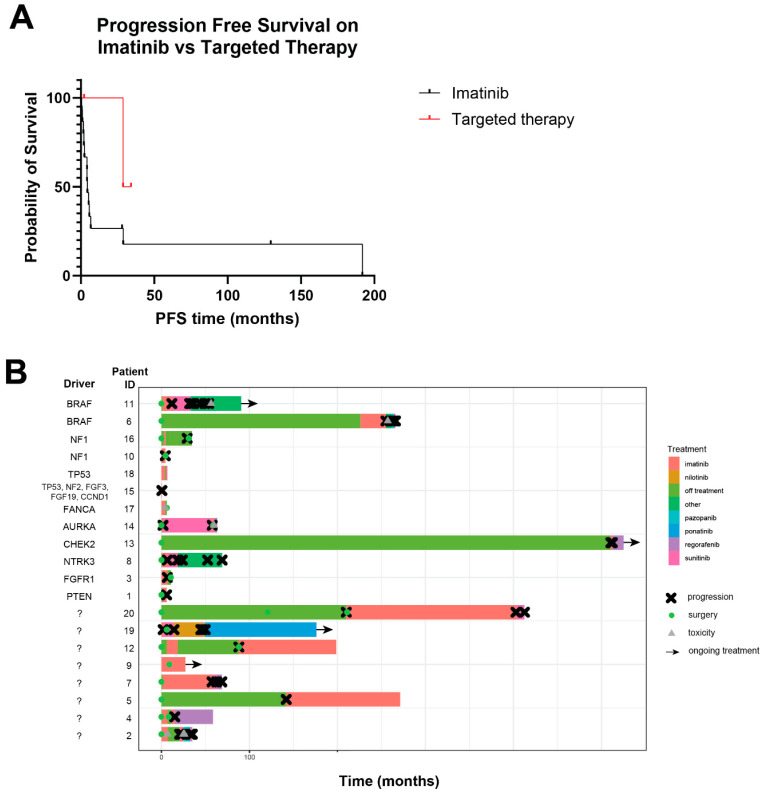
Response to treatment in triple-negative GIST. (**A**) Progression-free survival while on imatinib (n = 15 patients) versus molecularly matched treatments (n = 3 patients). *p* = 0.21. (**B**) Swimmer’s plot showing timeline of indicated therapies. Purported driver mutation is shown in the column on the left.

**Figure 4 cancers-16-01707-f004:**
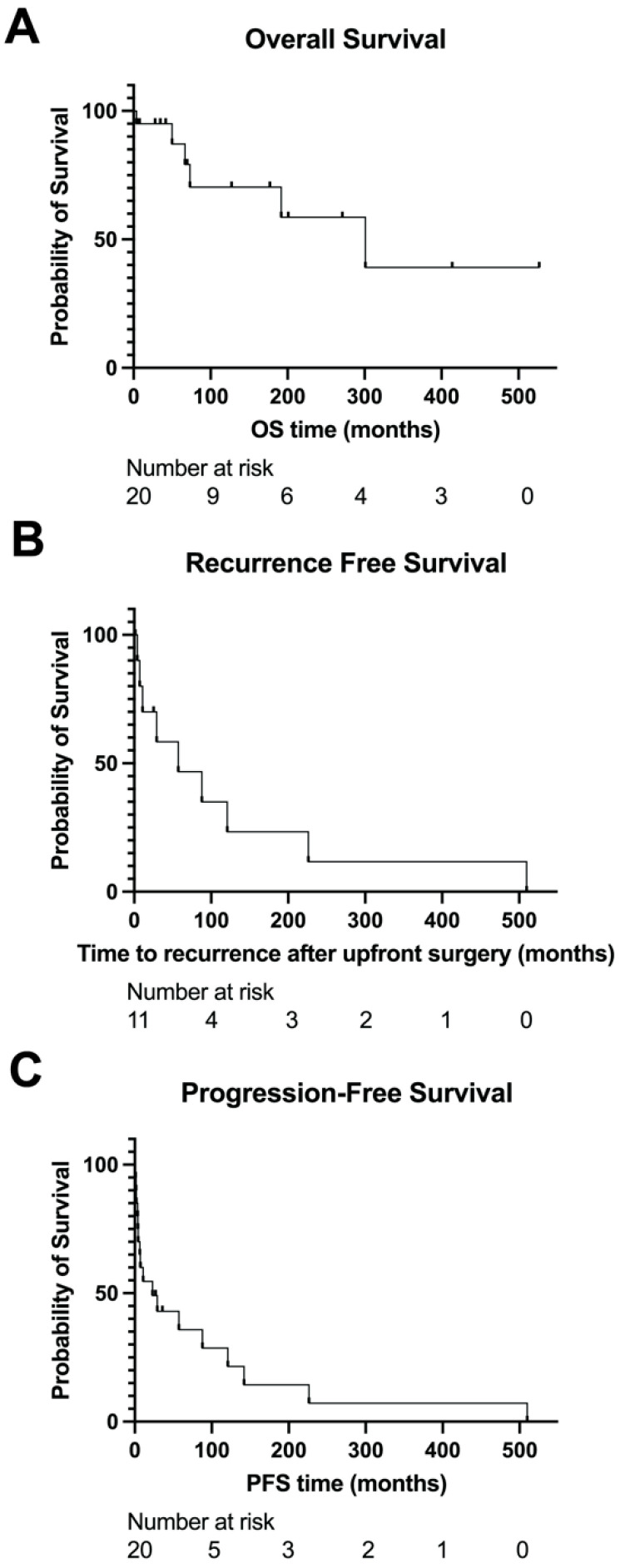
Survival outcomes in triple-negative GIST. Overall survival (**A**), recurrence-free survival (**B**), and progression-free survival (**C**) of patients with triple- negative GIST. Recurrence-free survival was calculated in patients with initially localized disease from the date of histologic diagnosis to the date of recurrence, death, or the latest follow-up. Progression-free survival was calculated from the start of therapy to the date of recurrence, death, or the latest follow-up.

**Table 1 cancers-16-01707-t001:** Clinical characteristics of triple-negative GIST cohort.

ID	Age at Diagnosis	Primary Site	Stage at Diagnosis (AJCC 8th)	Site of Recurrence/Metastasis	Imatinib Received in Which Setting?	PFS on Imatinib (Months)	Time to Relapse from First Therapy (Months)	Driver	Sequencing Platform
1	41	Gastric	IIIB	Local	Adjuvant	NA	4.1	*PTEN* deletion	Neotype, Boston Gene
2	68	Small intestine	IV	Pelvis	Adjuvant	NA	2.4	?	MDA
3	38	Small intestine	IA	NA	Neoadjuvant	6.7	NA	*FGFR1* K654E	MDA
4	21	Gastric	IB	Local, liver	Neoadjuvant and adjuvant	4.1	7.1	?	Boston Gene, Foundation One, MDA
5	17	Gastric	IV	Liver	Recurrent/metastatic	129.4	141.8	?	MDA
6	12	Gastric	?	Right psoas, retroperitoneum	Recurrent/metastatic	28.8	226.2	*BRAF* V600E	MDA
7	57	Small intestine	IB	Liver	Adjuvant	NA	57.4	?	MDA
8	44	Gastric	IV	Local, liver, spleen	Recurrent/metastatic	5.7	6.6	*ETV6-NTRK3* fusion	Foundation One, MDA
9	37	Gastric	IV	Omentum	Neoadjuvant	28	NA	?	OSI
10	72	Small intestine	IV	Pancreas	Neoadjuvant	5.2	4.4	*NF1* deletion	MDA
11	58	Small intestine	IIIB	Bladder, abdomen	Adjuvant	NA	10.8	*BRAF* V600E	MDA
12	48	Small intestine	IIIA	Local	Adjuvant	NA	88	?	MDA
13	14	Gastric	?	Liver, lung	Recurrent/metastatic	1.7	509.9	*CHEK2* frameshift	Foundation One
14	73	Small intestine	IV	Omentum, lymph nodes	Recurrent/metastatic	1	1	*AURKA-CSTF1* fusion	Foundation One
15	50	Gastric	?	Local, liver, pancreas, periportal region, duodenum, omentum	Neoadjuvant	0.5	2.2	*NF2* splice site mutation, *FGF3/FGF19* amplification, *CCND1* amplification, *TP53* frameshift	MDA
16	58	Gastric	II	Local	Adjuvant	NA	29.2	*NF1* splice site mutation	MDA
17	54	Rectum	IIIA	Peritoneum, ovary	Neoadjuvant	4.4	4.4	*FANCA* loss	Boston Gene
18	53	Small intestine	IV	Liver	Recurrent/metastatic	4.2	4.2	*TP53* missense and deletion	Caris, Boston Gene
19	25	Gastric	IV	Liver	Recurrent/metastatic	2.0	2.0	?	OSI panel
20	11	Gastric	?	Liver, omentum	Recurrent/metastatic	191.8	120.7	?	Endeavor, Boston Gene

MDA = MD Anderson in-house test; OSI = outside institution; ? indicates that no potential driver alteration was identified.

## Data Availability

De-identified data generated in this study are available upon request from the corresponding author.

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
