# Peer review of "Utility of Clinical Next Generation Sequencing Tests in KIT/PDGFRA/SDH Wild-Type Gastrointestinal Stromal Tumors"

_cancers, 2024, doi:10.3390/cancers16091707_

Round 1

Reviewer 1 Report

Comments and Suggestions for Authors

Authors reported the mutational profile of 20 cases of KIT/PDGFRA/SDH WT GISTs.

Different sequencing panels were used. Mutations found are in line with literature. These two limits the novelty of the paper. 

However since KIT/PDGFRA/SDH WT are extremely rare, the addition of 20 cases to the already sequenced in literature, it can be usefull and of some interest for the reader.

I have some minor comments:

- Different sequencing panels were used, each with some difference in the list of genes targeted. How authors hadle this difference? were only common genes considered? were the  mutations shown in figure 2A all covered by the assays used, or for some patients it must be indicated that the gene was not covered by the assay?

I have this doubt since i see that the patients tested with Foundation 1 are the ones with more alteration per sample in Fig2A.

- could authors provide the total number of somatic alteration identified in each samples? 

- in literature it has been reported the gain of FGF4 as a frequent event in quadruple WT GIST, in particular it was reported in 6 cases (https://doi.org/10.1002/gcc.22753  and https://doi.org/10.1038/s41598-020-76519-y). In the present paper the amplification of FGF3 and FGF19 was reported. Since these genes are upstream and downstream of FGF4 it can be  the same event reported in literature, providing one of the first confirmation on a different cohort of this finding. Can authors comment on that? Does MDA sequencing method provide information on FGF4 too?  It is a real amplification (more than 4 copies) or a gain (CN=3) ? Is this locus covered by the different assays used in the present paper or only by MDA method? 

-The review of literature it can be improved. the methods used for paper retrieval is not clear. For example they have cited some literature of Pantaleo MA, but hey have not used this paper cohort https://doi.org/10.1158/1541-7786.MCR-16-0376, which is one of the Pantaleo's biggest quadruple-WT cohort analyzed. Why?

Author Response

Authors reported the mutational profile of 20 cases of KIT/PDGFRA/SDH WT GISTs.

Different sequencing panels were used. Mutations found are in line with literature. These two limits the novelty of the paper.

However since KIT/PDGFRA/SDH WT are extremely rare, the addition of 20 cases to the already sequenced in literature, it can be usefull and of some interest for the reader.

I have some minor comments:

- Different sequencing panels were used, each with some difference in the list of genes targeted. How authors hadle this difference? were only common genes considered?

The fact that different sequencing panels were used is a limitation of the study, and we mention this in the final paragraph of the Discussion section as follows:

“Second, we are bound by the limitations of the sequencing panels to detect driver mutations, and the panels were not completely overlapping, though common drivers identified were included in all the panels.”

were the  mutations shown in figure 2A all covered by the assays used, or for some patients it must be indicated that the gene was not covered by the assay? I have this doubt since i see that the patients tested with Foundation 1 are the ones with more alteration per sample in Fig2A.

The reviewer brings up a great point. We have modified Fig 2A to add gray boxes to genes that were not covered in sequencing assays that were performed for each patient’s tumor.

- could authors provide the total number of somatic alteration identified in each samples? 

            We now provide a graph of the number of somatic mutations per tumor in Figure 2B.

- in literature it has been reported the gain of FGF4 as a frequent event in quadruple WT GIST, in particular it was reported in 6 cases (https://doi.org/10.1002/gcc.22753  and https://doi.org/10.1038/s41598-020-76519-y). In the present paper the amplification of FGF3 and FGF19 was reported. Since these genes are upstream and downstream of FGF4 it can be  the same event reported in literature, providing one of the first confirmation on a different cohort of this finding. Can authors comment on that? Does MDA sequencing method provide information on FGF4 too?  It is a real amplification (more than 4 copies) or a gain (CN=3) ? Is this locus covered by the different assays used in the present paper or only by MDA method? 

We thank the reviewer for pointing this out. The MDA sequencing method used for this patient did not include FGF4. The FGF3 and FGF19 amplifications are indeed real amplifications with copy number values of 16.8 and 16.75, respectively. This patient (patient 15) has several additional alterations that could be drivers: FGF3 and FGF19 amplification, CCND1 amplification, NF2 splice site mutation, and TP53 frameshift mutation. We now comment on FGFR1 pathway alterations as a driver of triple negative GIST in the Discussion as follows (pg 15-16):

“Outside of RAS/RAF/MAPK pathway alterations, one of the next most commonly altered pathways is the FGFR1 pathway [35]. Multiple studies have identified different alterations in this pathway. Some studies have identified GOF mutations in the FGFR1 receptor itself [36,37], including the current study. FGFR1 fusions have also been identified [37]. FGF3, FGF4, and FGF19 are adjacent genes on chromosome 11q13, are frequently co-amplified, and are known to be ligands for FGFR1 [38]. FGF4 amplification has been reported in wild type GIST [39], and our study identified FGF3 and FGF19 amplification in one case. Taken together, there appear to be multiple mechanisms leading to increased FGFR1 activity as a driver in a sizable fraction of quadruple negative GISTs. Interestingly, SDH-deficient GISTs have also been shown to upregulate FGF4 expression via DNA hypermethylation-mediated disruption of an insulator that normally separates a super-enhancer from the FGF4 gene [40]. Concordantly, an SDH-deficient patient-derived xenograft model was sensitive to FGFR inhibition [40]. Interestingly, regorafenib is a TKI approved for GIST that also targets FGFR and has shown significant benefit in SDH-deficient GIST [41]. Therefore, the FGFR pathway appears to be important for the pathogenesis of wild type GIST and may be an important therapeutic target.”

-The review of literature it can be improved. the methods used for paper retrieval is not clear. For example they have cited some literature of Pantaleo MA, but hey have not used this paper cohort https://doi.org/10.1158/1541-7786.MCR-16-0376, which is one of the Pantaleo's biggest quadruple-WT cohort analyzed. Why?

We thank the reviewer for this suggestion and apologize for this omission. We have now added this reference to the paper and the cases to Supplemental Table S1. In re-reviewing the literature, we also identified additional cases of triple negative GIST and have added these to Supplemental Table S1. We have accordingly edited the section “Review of literature on triple negative GIST” as follows:

Lastly, we reviewed the literature for all cases of triple negative GIST (Supplemental Table S1). Including the patients reported in our cohort, we identified a total of 112 cases. The mean and median age of diagnosis were 54.7 and 56.5 years, respectively. There were 49 females (51.0% of cases with sex reported) and 47 males (48.9%). There was a predilection for small intestinal GIST (64.9% of cases with primary site listed) compared to gastric (22.3%), colorectal (4.3%), and peritoneal/retroperitoneal (5.3%). The mean and median tumor size were 7.4 cm and 6.5 cm, respectively. The mean and median mitotic rate were 23.1 and 8 per 50 hpf or 5 mm2 (range <5 to 160), respectively. BRAF mutations (33 cases, 29.5%), NF1 LOF mutations (24 cases, 21.4%), and FGFR1 pathway alterations (13 cases, 11.6%) were the most common alterations. Other reported alterations in more than one case included NTRK3 fusion and TP53 LOF mutation.

Reviewer 2 Report

Comments and Suggestions for Authors

The article analyses stromal tumours and the importance of next-generation sequencing. The article title directs the reader to an expectation not covered in the manuscript, so I suggest partially changing it: The relevance of. Even though the article's data is analysed according to the major aim, there are questions. Please specify in the legend of Table 1 why there is no diagnosis in some patients even though the tumour was sequenced. Figures 2B and 2C may mislead the reader. I would suggest moving Figure 2C to a new Figure since the focus is on altered genes. Figure 3 is the most relevant; however, part B would benefit from clustering the patients' data with question marks. It would be clinically pertinent to direct the reader to the relevance of next-generation sequencing on patients treated with imatinib. Can the authors please provide the statistical analysis 3A and 4? Also, there are some limitations based on the data. It should be stated as a separate title in the text. Two interesting questions may be addressed: TP53 and RAS (not observed in Figure 2A) may affect stromal tumours, and only one patient with a complex clinical condition had TP53. What is the opinion of the authors? How about tumour heterogeneity in the samples? 

Minor grammatical mistakes were encountered

Comments on the Quality of English Language

Minor grammatical mistakes were encountered

Author Response

The article analyses stromal tumours and the importance of next-generation sequencing. The article title directs the reader to an expectation not covered in the manuscript, so I suggest partially changing it: The relevance of. Even though the article's data is analysed according to the major aim, there are questions.

To reflect the reviewer’s feedback, we have changed the title to:

“Utility of clinical next generation sequencing tests in KIT/PDGFRA/SDH wild-type gastrointestinal stromal tumors.”

Please specify in the legend of Table 1 why there is no diagnosis in some patients even though the tumour was sequenced.

The reviewer highlights an important point that despite broad clinical sequencing, we were unable to identify the driver for a number of the tumors. We have commented on why there is no diagnosis in the discussion as follows (pg 18):

Second, we are bound by the limitations of the sequencing panels to detect driver mutations, and the panels were not completely overlapping, though common drivers identified were included in all the panels.

Figures 2B and 2C may mislead the reader. I would suggest moving Figure 2C to a new Figure since the focus is on altered genes.

We think it makes most sense to keep Figure 2C with the remainder of Figure 2, as all of these panels focus on altered genes.

Figure 3 is the most relevant; however, part B would benefit from clustering the patients' data with question marks. It would be clinically pertinent to direct the reader to the relevance of next-generation sequencing on patients treated with imatinib.

            We have modified Figure 3B so that it is clustered by purported driver.

Can the authors please provide the statistical analysis 3A and 4?

We have provided a p value from a log rank test for Figure 3A in the legend (p = 0.21), though this is limited by small n (n=3 for targeted therapy and n = 15 for imatinib).

Also, there are some limitations based on the data. It should be stated as a separate title in the text. Two interesting questions may be addressed: TP53 and RAS (not observed in Figure 2A) may affect stromal tumours, and only one patient with a complex clinical condition had TP53. What is the opinion of the authors? How about tumour heterogeneity in the samples?

We did see TP53 alteration in 2 tumors in our cohort. We do not see RAS mutations; however, we did see RAS pathway alterations, namely 2 tumors with BRAF V600E and 2 with NF1 loss-of-function mutations. The reviewer also raises an interesting point about tumor heterogeneity. Unfortunately our study is not well suited to address intratumoral heterogeneity given that we only have sequencing results from one tumor biopsy/specimen from one point in time.

Minor grammatical mistakes were encountered

We have again reviewed the manuscript for grammatical errors and edited accordingly.

Reviewer 3 Report

Comments and Suggestions for Authors

This study focuses on "triple negative" gastrointestinal stromal tumors that lack mutations in KIT, PDGFRA, and SDH, aiming to elucidate their clinical and genomic characteristics. It uncovers that these GISTs harbor a diverse range of driver mutations, with patients showing a more favorable response to molecularly targeted therapies over imatinib, underscoring the importance of in-depth molecular profiling for guiding treatment. The publication is well-conducted, comprehensive, and leaves no room for critique in its execution or findings.

Author Response

We thank the reviewer for this kind feedback.